# Patients with Advanced or Metastasised Non-Small-Cell Lung Cancer with *Viscum album* L. Therapy in Addition to PD-1/PD-L1 Blockade: A Real-World Data Study

**DOI:** 10.3390/cancers16081609

**Published:** 2024-04-22

**Authors:** Friedemann Schad, Anja Thronicke, Ralf-Dieter Hofheinz, Harald Matthes, Christian Grah

**Affiliations:** 1Research Institute Havelhöhe, Network Oncology Registry, Kladower Damm 221, 14089 Berlin, Germany; 2Hospital Gemeinschaftskrankenhaus Havelhöhe, Interdisciplinary Oncological Centre, Kladower Damm 221, 14089 Berlin, Germany; 3Mannheim University Hospital, Mannheim Cancer Center, Theodor-Kutzer Ufer 1-3, 68167 Mannheim, Germany; 4Charité-Universitätsmedizin Berlin, Corporate Member of Freie Universität Berlin and Humboldt-Universität zu Berlin, Department of Gastroenterology, Hindenburgdamm 30, 12203 Berlin, Germany; 5Hospital Gemeinschaftskrankenhaus Havelhöhe, Daycare Clinic, Kladower Damm 221, 14089 Berlin, Germany; 6Hospital Gemeinschaftskrankenhaus Havelhöhe, Lung Cancer Center, Kladower Damm 221, 14089 Berlin, Germany; christian.grah@havelhoehe.de

**Keywords:** PD-1 inhibitor, PD-L1 inhibitor, survival, *Viscum album* L. extracts, non-small-cell lung cancer, lung cancer, real-world data study

## Abstract

**Simple Summary:**

Immunotherapy with PD-1/PD-L1 inhibitors has been shown to significantly improve the survival rates of non-small-cell lung cancer (NSCLC) patients in advanced or metastasised stages. In the present real-world data study, we examined patients with advanced or metastasised NSCLC and compared overall survival between a control group receiving a PD-1/PD-L1 inhibitor therapy and the combinatorial group receiving PD-1/PD-L1 inhibitors plus add-on *Viscum album* L. Our findings revealed that patients in the combinatorial group lived seven months longer in comparison to the control group (*p* < 0.001). Furthermore, in patients with a PD-L1-positive tumour that received first-line anti-PD-1/PD-L1 therapy, the adjusted hazard of death was reduced by 56% through the addition of *Viscum album* L. (*p* < 0.001). Our findings imply that the addition of *Viscum album* L. to PD-1/PD-L1 inhibitors is associated with improved survival in patients with advanced or metastasised NSCLC. A limitation of this study is the observational and non-randomised study design; prospective randomised trials are warranted.

**Abstract:**

Immunotherapy with PD-1/PD-L1 inhibitors has significantly improved the survival rates of patients with metastatic non-small-cell lung cancer (NSCLC). Results of a real-world data study investigating add-on VA (*Viscum album* L.) to chemotherapy have shown an association with the improved overall survival of patients with NSCLC. We sought to investigate whether the addition of VA to PD-1/PD-L1 inhibitors in patients with advanced or metastasised NSCLC would have an additional survival benefit. In the present real-world data study, we enrolled patients from the accredited national registry, Network Oncology, with advanced or metastasised NSCLC. The reporting of data was performed in accordance with the ESMO-GROW criteria for the optimal reporting of oncological real-world evidence (RWE) studies. Overall survival was compared between patients receiving PD-1/PD-L1 inhibitor therapy (control, CTRL group) versus the combination of anti-PD-1/PD-L1 therapy and VA (combination, COMB group). An adjusted multivariate Cox proportional hazard analysis was performed to investigate variables associated with survival. From 31 July 2015 to 9 May 2023, 415 patients with a median age of 68 years and a male/female ratio of 1.2 were treated with anti-PD-1/PD-L1 therapy with or without add-on VA. Survival analyses included 222 (53.5%) patients within the CRTL group and 193 (46.5%) in the COMB group. Patients in the COMB group revealed a median survival of 13.8 months and patients in the CRTL group a median survival of 6.8 months (adjusted hazard ratio, aHR: 0.60, 95% CI: 0.43–0.85, *p* = 0.004) after adjustment for age, gender, tumour stage, BMI, ECOG status, oncological treatment, and PD-L1 tumour proportion score. A reduction in the adjusted hazard of death by 56% was seen with the addition of VA (aHR 0.44, 95% CI: 0.26–0.74, *p* = 0.002) in patients with PD-L1-positive tumours (tumour proportion score > 1%) treated with first-line anti-PD-1/PD-L1 therapy. Our findings suggest that add-on VA correlates with improved survival in patients with advanced or metastasised NSCLC who were treated with PD-1/PD-L1 inhibitors irrespective of age, gender, tumour stage, or oncological treatment. The underlying mechanisms may include the synergistic modulation of the immune response. A limitation of this study is the observational non-randomised study design, which only allows limited conclusions to be drawn and prospective randomised trials are warranted.

## 1. Introduction

Lung cancer—ranking second among all cancers—remains the main cause of cancer deaths and is among the top five causes of death worldwide with NSCLC being the most common subtype of lung cancer [1]. In 2020, more than two million people were diagnosed with lung cancer worldwide. While the 5-year survival rate for general NSCLC is currently 28% (33% in women and 23% in men), the 5-year survival rate of metastatic NSCLC is 9% with higher survival rates in those patients receiving targeted or immunotherapies [1,2].

Immune checkpoint blockade (ICB) as a type of immunotherapy aims to unleash the immune system of the body to identify and combat cancer cells. ICB has shown promising results in the treatment of various types of cancers, with NSCLC among them. It works through the targeting of cytotoxic T-lymphocyte antigen-4 (CTLA-4), programmed cell death protein 1 (PD-1), or programmed death-ligand 1 (PD-L1), known as immune checkpoints, which inhibit immune response [3,4] and can be employed by cancer cells to escape attacks from the immune system.

Recent advances in ICB research and development have significantly improved options and outcomes for NSCLC patients including (a) the approval of ICB as first-line combinational or monotherapy in stage IV NSCLC without driver alterations depending on PD-L1 expression [5,6,7,8] or as an adjuvant therapy in non-metastatic NSCLC for certain conditions [9,10]; (b) the enhanced biomarker development including tumour mutational burden and microsatellite instability [11]; (c) ICB’s combination with other targeted, immuno-, chemo- or radiotherapy [8]; (d) the exploration and development of novel immune checkpoint inhibitors, multispecific antibodies, and other immunomodulatory agents as well as the exploration of new immune regulation pathways [12,13]. To maximise the efficacy and minimise the toxicity of ICB or the development of resistance, ongoing research is being continued for the optimisation of immunotherapy approaches including personalised therapy strategies and research on immune evasion pathways, tumour microenvironment factors and host immune response [2,14,15].

A number of immune checkpoint inhibitors have already received approval in Europe for first-line therapies in the field of advanced or metastasised NSCLCs [16,17,18,19,20,21]. Pembrolizumab is approved as (a) an adjuvant treatment after complete resection and platinum-based chemotherapy in NSCLC patients with a high recurrence prognosis; (b) a first-line therapy in metastatic high-PD-L1 (≥50% TPS) NSCLC without EGFR mutations or ALK translocations; (c) a first-line PD-L1 status-independent therapy combined with platinum-based chemotherapy in metastatic squamous NSCLC; (d) a first-line PD-L1 status-independent treatment in combination with platinum-based chemotherapy in metastatic non-squamous NSCLC with no EGFR or ALK aberrations; and (e) a second-line therapy of locally advanced or metastatic low-PD-L1 (≥1%)-expressing NSCLC following chemotherapy in patients with EGFR or ALK aberrations following targeted therapy [16]. Nivolumab is approved as a first-line therapy in combination with ipilimumab and two cycles of platinum-based chemotherapy in metastatic NSCLC without EGFR or ALK aberrations [18]. In addition, nivolumab can be given as a monotherapy in locally advanced or metastasised NSCLC after chemotherapy or in the neo-adjuvant setting in resectable, low-PD-L1 (≥1%) NSCLC [18]. Cemiplimab is approved in Europe as a first-line therapy of locally advanced or metastatic low-(≥1%) or high-PD-L1 (≥50% TPS) NSCLC without EGFR, ALK, or ROS aberrations and only when not indicated for definitive radiochemotherapy [19]. Atezolizumab is approved in Europe as a first-line therapy in metastatic NSCLC with no EGFR or ALK aberrations (a) in tumours with a high (≥50%) PD-L1 status or ≥10% tumour-infiltrating cells; (b) as a combinational therapy with nab paclitaxel and carboplatin in non-squamous tumours; or (c) as a combinational therapy with bevacizumab, paclitaxel, or carboplatin in non-squamous tumours. The utilisation of the latter combination is allowed as well in EGFR- or ALK-positive non-squamous NSCLC but only after the failure of targeted therapy [20]. Lastly, durvalumab is approved in Europe as a first-line therapy combined with tremelimumab and platinum-based chemotherapy in stage IV NSCLC without EGFR or ALK aberrations. In addition, durvalumab can be applied in non-resectable advanced low-PD-L1 (≥1%) NSCLC without progression after platinum-based radiochemotherapy [21]. Thus, the first-line application of anti-PD-1/PD-L1 treatments has developed broadly and has led to important improvements in the survival rates and health-related quality of life of the patients during the recent decade reflecting the grand efforts in finding a more effective therapy for this devastating disease.

*Viscum album* L. (European white-berry mistletoe, VA) is approved in Germany for subcutaneous adjuvant and palliative treatment in malignant tumour diseases [22,23,24,25] and systematic reviews and meta-analyses reveal a significant medium-size effect in the improvement of health-related quality of life and fatigue [26,27,28,29,30]. In addition, add-on VA has been associated in various studies and clinical studies with the improvement of survival [31,32,33,34,35]. Hereby, the anti-proliferative [36], pro-apoptotic [37], immunomodulatory [38,39,40,41,42], anti-nociceptive, and anti-depressant [43] properties of the VA extracts seem to play a role. Two recent systematic reviews and meta-analyses revealed improved survival in oncological patients treated with add-on VA therapy—the reduction in hazard of death ranged from 19% (HR 0.81, 95% CI: 0.69–0.95, *p* = 0.01) to 41% (HR 0.59, 95% CI: 0.53–0.65, *p* < 0.0001) [35,44]. Furthermore, VA in addition to standard oncological chemotherapy revealed to be associated with improved overall survival in advanced or metastasised NSCLC patients [31,45,46,47]. So far, the association of combined PD-1/PD-L1 inhibitor plus add-on VA therapy with survival outcomes of advanced or metastasised NSCLC patients has not been studied systematically. First real-world evidence (RWE) results documented no enhanced adverse effects when VA was given in addition to PD-1/PD-L1 inhibitor therapy in patients with advanced or metastasised melanoma or lung cancer [48,49,50,51]. Furthermore, in another small RWE study with advanced or stage IV lung cancer patients, the PD-1 inhibitor-induced toxicity was almost halved when VA was added [49].

The aim of our study was to investigate the overall survival of advanced or metastasised NSCLC patients receiving ICB as outlined above with or without VA.

## 2. Materials and Methods

### 2.1. Study Design, Description of Study Participants, and Data Source Assessment

The present study is a real-world data study evaluating data from oncological patients registered in the clinical registry, Network Oncology [33], which is accredited by the German Cancer Society. The study was retrospectively registered on 27 November 2017 (trial registration number DRKS00013335). The primary aim was to investigate the overall survival of patients with advanced or metastasised NSCLC receiving PD-1/PD-L1 inhibitors with or without VA. The secondary aim was to evaluate whether certain tumour or patient characteristics were associated with the reduction in the hazard of death. Patients with advanced or metastasised (UICC stages III–IV) NSCLC that have received PD-1/PD-L1 inhibitor therapy with or without VA therapy registered in the Network Oncology were included in the analysis. Further inclusion criteria were patients being eighteen years or older, of male or female gender, and who have given their written consent. Patients’ demographic data; date and type of histological tumour confirmation; Union Internationale Contre Le Cancer (UICC) tumour stage (TNM classification, 8th edition, the UICC TNM classification of the tumour as an internationally accepted standard for cancer staging); Eastern Cooperative Oncology Group (ECOG) stage; body mass index (BMI); smoking status; comorbidity status (occurrence or absence); oncological treatment; and survival state, death date, or tumour conference or last contact date were extracted from the Network Oncology registry were retrieved. The information on applied VA extracts with start and end dates being given in the context of an integrative oncological setting was retrieved as well. In accordance with the NO requirements, the patients were followed up routinely six months after the first diagnosis and annually during the next years [52]. A loss to follow-up was defined when no follow-up visits were documented.

### 2.2. Ethics Approval and Consent to Participate

The study has been approved by the ethics committee of the Medical Association Berlin (Eth-27/10). Written informed consent was obtained from all patients prior to their enrolment in the study. The study complies with the principles laid out in the Declaration of Helsinki.

### 2.3. Group Classification

Patients were classified into the following histological subgroups: squamous cell carcinoma, non-squamous cell carcinoma, or large-cell carcinoma. After that, patients were classified into one of two groups: either (a) the control (CTRL) group where the patients received PD-1/PD-L1 inhibitors and no VA therapy or (b) the combinational (COMB) group where they received PD-1/PD-L1 inhibitors and an add-on VA therapy. Add-on VA was given at the discretion of the doctor and the decision of the patient after elaborate consultation on the patient’s treatment options. The allocation of the patient to one of the treatment groups was performed non-randomly. Both, PD-1/PD-L1 inhibitors or the combination with VA were applied as routine clinical care. Patients received add-on VA therapy subcutaneously in line with the summary of product characteristics (SmPC) [23,24,25]. Off-label intravenous or intratumoural application was performed in individual cases. Applied VA preparations included VA from Helixor GmbH, Rosenfeld, Germany, Abnoba GmbH, Niefern-Öschelbronn, Germany and/or Iscador AG, Arlesheim, Switzerland.

### 2.4. Statistical Methods

Continuous variables were summarised as medians with an interquartile range (IQR) while categorical variables were summarised in absolute and relative frequencies. All analyses were explorative. The distributions of the data were checked graphically and arithmetically. Patients with missing data were not included. The comparison of baseline characteristics and treatment regimens was performed with the unpaired Student’s *t*-test (independent normally distributed samples) or the Mann–Whitney U test (non-normally distributed data). A chi-square analysis with Yates’s continuity correction was used when comparing categorical variables. Tests were performed two-sided. In the survival analyses, the start date (index date) was the first date of the start of anti-PD-1/PD-L1 therapy. Patient survival data were then analysed from the index date until the patient’s last record, which was either the date of death, the date of the interdisciplinary tumour board conference, the last documentation date of the personal contact with the patient, or the follow-up date (for follow-up measures, please see Study Design and patients,). Patients were censored when alive at the time of the analysis. Kaplan–Meier survival was calculated for both the CTRL and COMB groups.

To analyse variables that were associated with the hazard of death and for the reduction in potential confounders, a multivariate stratified Cox proportional hazard analysis was performed adjusting for age, gender, ECOG (Eastern Europe Oncology Group) status, PD-L1 status, BMI, smoking status, comorbidity status, and oncological treatment. Before that, we performed verification analyses to check whether the proportional hazard was met. The analyses were performed with the software R, 4.1.2 (1 November 2021) “bird hippie”, R-Studio version 2022.02.2, a language and environment for statistical computing [53]. Kaplan–Meier survival analyses and Cox proportional hazard analyses were performed using the R-package ‘survival’, version 3.5-5 [54]. For the implementation of nonparametric estimators for censored event history (survival) analysis, the package ‘prodlim’ was applied, version 2019.11.13 [55]. To draw survival curves, the package ‘survminer’ was used, version 0.4.9.

### 2.5. Reporting of RWE Data

We reported our real-world evidence data in accordance with the ESMO guidance for optimal reporting of oncology real-world evidence (GROW) [56]. According to the ESMO-GROW checklist for this study, which is submitted alongside our manuscript, the ESMO-GROW informative score is 27.0 points out of 29 (93%, see Appendix A).

## 3. Results

### 3.1. Baseline Characteristics

Four hundred and fifteen (*n* = 415) patients with advanced or metastasised NSCLC receiving PD-1/PD-L1 inhibitors as the standard of care of the Network Oncology registry were enrolled.

Two hundred and twenty-two patients (53.5%) were treated with PD-1/PD-L1 inhibitors without additional VA therapy (CTRL) and one hundred and ninety-three patients (46.5%) with PD-1/PD-L1 inhibitors and additional VA therapy (COMB) (see the flowchart in Figure 1).

We observed no significant differences when comparing both groups as to gender, histology, smoking status, comorbidities, ECOG status, or tumour stage; see Table 1. The median age of the study group was 68 years. The sex ratio (male/female) was 1.2. Most participants had an ECOG status of 1 (42.7%) followed by ECOG 0 (25.5%) and ECOG 2 (12.8%); see Table 1. The most prevalent NSCLC histology was non-squamous cell carcinoma with 65.3% (*n* = 271), followed by squamous cell carcinoma with 27.2% (*n* = 113); see Table 1. In 7.5% of the patients (*n* = 31), the NSCLC diagnosis was not histologically specified. The proportion of patients diagnosed with non-squamous cell carcinoma was 3.9% higher in the COMB group while the proportion of patients diagnosed with squamous cell carcinoma was 2.5% lower in the COMB group in comparison to the CRTL group; the differences between both groups were not significant. More than half of the patients were of normal weight (BMI below 25 kg/m^2^) while more than a third were overweight (BMI 25–29.9 kg/m^2^). When combining normal and overweight (25–29.9 kg/m^2^) patients and comparing them to obese patients (≥30 kg/m^2^), a significantly higher proportion of overweight patients was observed in the CTRL group (*p* = 0.04); see Table 1.

For sixty patients (14.5%), the smoking status was not documented. More than two-thirds of patients were current or past smokers with a slightly higher proportion in the COMB compared to the CTRL group, with no significant differences between the groups. More than two-thirds of the participants had a UICC stage IV tumour (78.3%) and 21.7% had stage III NSCLC. In total, 15 patients had an initial early-stage NSCLC that progressed to stage IV NSCLC, with 9 (4.1%) being in the CTRL group and 6 (3.1%) in the COMB group. When combining both groups, a non-significant trend for a higher proportion of stage IV NSCLC was observed in the CTRL group, see Table 1.

### 3.2. Molecular Markers

No significant differences in the molecular marker level between both groups were observed; see Table 2. While the PD-L1 status was known for 82.2% of all observed patients, the status of known EGFR (exon 18–21) mutations, ROS rearrangements, BRAF mutations, or ALK translocations ranged from 45.8% to 56.6%; see Table 2. As for stage IV NSCLC, for 268 (82.5%) of 325 patients, the molecular status was documented.

In Appendix A, the different EGFR (Exon 18–21) mutations were further classified into EGFR insertion, deletion, point, or duplication mutations. In total, two (0.5%) insertion mutations, five (1.2%) deletion mutations, six (1.4%) point mutations, and two (0.5%) duplication mutations were documented for the thirteen patients with EGFR mutations. The between-group comparison revealed no significant differences between both groups as to various mutation types; see Appendix A. Eight (1.9%) of the total observed EGFR mutations were known to show sensitivity against tyrosine kinase inhibitors (TKIs) while two other mutations revealed no sensitivity against TKIs or no sensitivity against TKIs of the 1st, 2nd, or 3rd generation. The between-group comparison revealed no significant differences between both groups as to sensitivity against TKIs; see Appendix A. The adjusted multivariable Cox regression analysis revealed a positive direction of the hazard of death for EGFR-mutant NSCLC (HR 1.07, 95% CI: 0.48–2.39, *p* = 0.87); see Appendix A.

For three hundred and forty-one patients (82.2%), the PD-L1 status was known, and in Table 3 the PD-L1 tumour proportion score has been categorised. No significant differences were seen between the groups. When comparing negative and positive PD-L1 tumour proportions between both groups, a non-significant trend towards a higher proportion of PD-L1-positive NSCLC in the CTRL group was observed; see Table 2.

### 3.3. Oncological Treatment

No significant differences in oncological treatment including chemotherapy, radiation, or PD-1/PD-L1 therapy were seen; see Table 4. As to antibody treatment, PD-1 inhibitors were the most applied immune checkpoint inhibitor therapy (95%) compared to PD-L1 inhibitors (5%); see Table 4. In the total cohort, 246 (59.3%) patients received first-line ICB while 169 (40.7%) received second-line ICB after targeted therapy or second-line anti-ICB. Add-on VA therapy started at the same time as the ICB in 82.4% of the patients and was started after ICB in 23.8% of the patients (median time to VA: 22 days).

### 3.4. Outcomes, Five-Year Survival

We included four hundred and fifteen patients in the Kaplan–Meier survival analysis. A significant survival improvement was seen for the COMB treatment group (PD-1/PD-L1 inhibitors + VA) compared to the CTRL group (PD-1/PD-L1 inhibitors without VA); see Figure 2 and Table 5.

The five-year survival analysis showed that the COMB group had a significant 7-month-longer median survival than the patients in the CTRL group; see Table 5 and Figure 2. The median survival was 13.8 months in the COMB group and 6.8 months in the CTLR group (χ^2^ = 17.5, *p* < 0.001); see Figure 2 and Table 5.

A multivariate Cox proportional hazard analysis adjusting for various factors revealed a statistically significant reduction in the hazard of death by 40% (adjusted hazard ratio—aHR: 0.60, 95% CI: 0.43–0.85, *p* = 0.004); see Table 6. This effect was independent of age, gender, metastasised stage, ECOG status, BMI, comorbidity status, smoker status, cranial radiation, surgery, or PD-L1 status. The adjusted multivariate analysis also revealed that having a comorbidity, an overweight or obese BMI, or female gender was associated with a reduced adjusted hazard while an increasing ECOG status was associated with an increased adjusted hazard. For PD-L1 TPS, a growing tendency towards a significant negative hazard of death was observed with growing TPS status; see Table 6.

### 3.5. Subgroup Analysis of First-Line PD-1/PD-L1 Therapy in Advanced or Metastasised NSCLC with Low PD-L1 Status

In a subgroup analysis (*n* = 171) of advanced or metastasised NSCLC patients with PD-L1 TPS ≥ 1% and first-line immune checkpoint blockade, the Kaplan–Meier survival analysis revealed a survival advantage when the combinational PD-1/PD-L1 inhibitor plus an add-on VA therapy was applied compared to the PD-1/PD-L1 inhibitor therapy alone; see Figure 3 and Table 7. ICB started in a median of 24 days (IQR 14–32 days) after NSCLC diagnosis in the subgroup.

The median OS in the COMB group (26.4 months) was 21 months longer than the median OS in the CTRL group (5.4 months), *p* < 0.001; see Table 7. The one-year survival rate in the COMB group was 66.3% and in the CTRL group, it was 34.4; meanwhile, the five-year survival in the COMB group was 16.5% compared to 5.7% in the CRTL group.

An adjusted Cox proportional hazard analysis revealed a 56% reduced adjusted hazard of death (aHR 0.44, 95% CI: 0.26–0.74, *p* = 0.002) when add-on VA was added to first-line PD-1/PD-L1 therapy in this subgroup with PD-L1-positive NSCLC; see Figure 4. The adjusted multivariate analysis also revealed an association between the occurrence of a comorbidity (reduced adjusted hazard, aHR, 0.53, 95% CI: 0.29–0.96) and an increasing ECOG status (increased adjusted hazard, aHR, 1.38, 95% CI: 1.09–1.74); see Figure 4.

## 4. Discussion

In the present study, we examined the efficacy of PD-1/PD-L1 inhibitor therapy in combination with add-on VA therapy in patients with advanced or metastatic NSCLC. Our findings indicate a significant association with survival benefits for these patients when treated with ICB in combination with VA therapy in comparison to ICB alone. Hereby, add-on VA reduced the adjusted hazard of death by 40%. Our findings comply with the results of two systematic reviews and meta-analyses stating a reduction in the general hazard of death in oncological patients after add-on VA therapy between 19% and 54% [35,44]. As to NSCLC patients, our findings mimic the effect of another RWD study in stage IV patients who were treated with standard chemotherapy and where add-on VA therapy reduced the hazard of death by 56% [31]. However, neither the above-mentioned systematic reviews nor the RWD study included anti-PD-1, -PD-L1, or -CLT-4A treatments for the evaluation of the association with survival. Thus, our study is the first of its kind, giving the first insights into this research.

The overall survival in the present study was 13.8 months in the combination group (ICB + add-on VA) compared to 6.8 months in the control group (ICB), and the difference was significant. When comparing our findings with published RWD studies involving ICB treatment in general, we found comparable results. One study reported a median overall survival of stage I-IV NSCLC patients of 9.3 months (95% CI 8.5–10.5 months) after the start of immune checkpoint inhibitors with 4.4% fewer ECOG ≥ 2 patients compared to our study [57]. Another RWD study observed an overall survival between 13.2 months (PD-L1 TPS < 1%) and 16.3 months (PD-L1 TPS 1–49%) in metastatic non-squamous NSCLC patients with an ECOG performance status from 0 to 1 who were treated with pembrolizumab plus pemetrexed–carboplatin [58].

Due to different patient cohort characteristics, RWD findings are not directly comparable to survival parameters of current randomised controlled trials (RCTs). Thus, the one-year survival rate of 68.8% in ICB treatment of the Keynote-042 RCT is higher compared to the one-year survival rate in our cohort (34.4%). This could be driven by the fact that our cohort more closely mirrors a real-world NSCLC cohort with an older patient population (68 years) in line with the median age of NSCLC patients [59] in the national cancer database (69 years) in comparison to the Keynote-042 trials (63 years) [60]. Furthermore, the proportion of patients with a histological squamous cell cancer type in our study, 27.2%, mirrors the real-world situation of 28% [59] better than the 55% recorded in the Keynote-042 China trial [61]. Last but not least, our cohort included patients with all ECOG levels compared to the Keynote-042, where only patients with ECOG 0-1 were included [60]. In one of the first head-to-head comparison studies between RCT and RWE studies, shorter OS has been observed in RWE studies for real-world first-line pembrolizumab therapy in stage IV NSCLC, partly due to a higher number of patients with performance status ≥ 2 in RWE studies [62]. Therefore, the findings of our study may reveal a real-world situation for NSCLC patients and do not reflect eligibility procedures in RCTs using strict inclusion criteria. In addition, our RWE study included first- as well as second-line ICB treatment. All mentioned factors may, therefore, account for the poorer observed curves in our study in general.

The multivariable regression analysis in our study indicated that the additional VA therapy was associated with a significant reduction in the adjusted hazard of death in advanced or metastasised NSCLC patients treated with ICB. This effect was independent of age, gender, tumour stage, surgery, ECOG status, BMI, comorbidity status, smoking status, cranial irradiation, or PD-L1 TPS. However, when evaluating a subgroup where only NSCLC patients with PD-L1 TPS ≥ 1% and first-line ICB were included, the add-on VA was associated with a reduction in the adjusted hazard of death (56%) compared to the general cohort (40%) as well. Therefore, one might speculate that VA extracts may trigger or mediate a PD-L1-dependent or -mediated immune response in NSCLC patients. This hypothesis is supported by in vitro research where VA extracts downregulated PD-L1 expression in 3D spheroids of breast cancer cell lines T47D and HCC1937 [63]. At the same time, no negative effect of VA therapy on the effect of atezolizumab or pembrolizumab was observed [63]. These in vitro data suggest that VA extracts are involved in the mediation of PD-L1-regulated processes in cancer cells. However, these data need further confirmatory analyses [63].

In vitro and in vivo studies revealed that the escape of potentially immunogenic tumours from the immune response of the host is mainly driven by the activation of the PD-1/PD-L1 signalling pathway [64,65,66]. While PD-1 was already known in the early 1990s as being expressed in lieu of apoptosis induction [66], it was later identified as a co-inhibitory molecule for T-cell activation, being involved in the negative regulation of apoptotic immunological reactions by binding to PD-L1 [67]. While PD-L1 binding to PD-1 leads to impaired anti-tumour activity, it was shown for lung cancer that this involved the activation of the AKT/ß-catenin/WIP signalling pathway [68]. Some studies suggest that VA extracts may also inhibit phosphoinositide 3-kinase (P13K), thereby affecting the downstream AKT signalling [69,70,71,72,73]. The AKT pathway is involved in the regulation of apoptosis; thus, VA’s effects on this pathway could be attributed to its pro-apoptotic properties. Therefore, combining immune checkpoint inhibitors and VA could potentially synergistically target identical or multiple checkpoints and immune regulatory pathways, leading to a comprehensive and effective, i.e., stronger, inhibition of tumour growth. But beyond these hypotheses, the specific mode of action concerning how both treatment modalities act to increase survival of advanced or metastasised NSCLC patients still needs to be elucidated further.

We were unable to consider the association between EGFR mutations and overall survival in our study because only a small number of patients had EGFR mutations. However, our adjusted multivariable Cox regression analysis revealed a positive direction of the hazard of death, i.e., the probability of death is increased in the presence of common EGFR mutations. The efficacy of ICB can be influenced by the expression and activity of immune-related molecules which in turn are influenced by driver aberrations such as EGFR, ALK, KRAS, or others. It has been shown that, e.g., in EGFR-mutant lung cancer, the EGFR signalling pathway is activated, leading likely to an uninflamed tumour microenvironment [74] due to the lowered activity of pro-inflammatory interleukins such as IL-6, IL-8, or TNF-α and due to the increased activity of immunosuppressive factors such as TGF-β, resulting in tumour escape or immunosuppression. Thus, these patients do not benefit from ICB therapy. TKI-treated NSCLCs with EGFR mutation often develop resistance which may limit therapy’s efficacy, mostly due to secondary mutations or compensatory pathways. The latter of the two adaptive mutability routes includes the activation of pharmacologically sensitive endogenous genes that enable continuous error-prone DNA replication in order to tolerate cancer therapy. Blocking involved genes or proteins may overcome these resistance mechanisms [75,76]. One of these proteins is the tyrosine kinase receptor Axl which activates intrinsic mutators and helps tumour cells evade tyrosine kinase inhibition [76]. The treatment of NSCLC cell lines with EGFR TKI resulted in Axl overexpression, the viability of tumour cells, and the inhibition of apoptotic pathways. It was observed in in vitro, in vivo experiments, and patient samples that Axl-mediated TKI induction of error-prone DNA polymerases and the acceleration of common EGFR resulted in drug resistance. In patient-derived xenografts, this effect was reversed through anti-Axl therapy [75]. The resistance to ICB is an emerging area in cancer therapy. Axl, which has been correlated to immune suppression, resistance to immunity, and lower response rates, was shown to be strongly associated with the PD-1 expression of the tumour. When treated with PD-1 inhibitors, tumours with high Axl expression revealed lower response rates and a trend toward shorter progression-free survival [77]. In addition, the activation of TAM receptors including Axl negatively modified the immune response, leading to an immunosuppressive and pro-tumorigenic tumour microenvironment, suggesting the combination of TAM receptor inhibitors and checkpoint inhibitors as long-term clinical therapeutic strategies in NSCLC [76,78]. Thus, combinational ICB strategies will play a pivotal role in the future to effectively reduce tumour cell survival and metastatic potential.

Lastly, in our study, we observed that an increased ECOG level was associated with an increased adjusted hazard of death while the occurrence of a co-morbidity at baseline was associated with a reduced adjusted hazard. The first association observed is in line with a systematic review and meta-analysis of real-world data covering data from 19 studies and 3600 NSCLC patients, showing that patients with a performance score ≥ 2 had a worse outcome in terms of OS, progression-free survival, and overall response rate [79]. The latter association in our study that the occurrence of a comorbidity is associated with a better survival outcome does not seem plausible at first sight. However, a recent real-world data study on 431 advanced cancer patients treated with PD-1/PD-L1 inhibitors revealed that (a) ≥grade 3/4 immune-related adverse events (irAEs) are affected by comorbidities including diabetes mellitus and others and that (b) patients with these ≥grade 3/4 irAEs revealed an improved progression-free survival (PFS) compared to patients without any adverse events [80]. This is in accordance with a recent systematic review and meta-analysis of 24 studies which showed that developing irAEs during PD-1/PD-L1 therapy was significantly associated with a 26% and 45% reduced hazard for OS and PFS, respectively, with endocrine, skin, gastrointestinal, and low-grade irAEs showing the best outcomes [81].

In the combinational setting of the present study, ICB with add-on VA was started in the first-line setting a median of 24 days after NSCLC diagnosis. Until now, no systematic research on the combinational ICB plus VA therapy and timing exists in the literature, with our data providing the first data on this topic so far. For other combinations including chemotherapy and add-on VA, findings of an RCT reported an application start of the combination 1 week after surgery with a duration of 23 weeks in patients with gastric cancer [82]. Another study, a real-world data study, revealed an effective application duration of add-on VA at a minimum of four weeks alongside chemotherapy in advanced or metastasised NSCLC patients, and the adjusted hazard was further reduced when add-on VA duration was prolonged to greater than sixteen weeks [31]. Add-on VA can be applied directly following a tumour diagnosis in an adjuvant setting during chemo- and/or radiochemotherapy, helping to improve patient health-related quality of life by reducing the adverse effects of standard oncological therapies [30,83,84].

Several systematic reviews and meta-analyses, clinical as well as real-world data studies, point towards the positive impact of VA extracts on survival [31,32,33,34,35,44,46,47], indicating that clinical findings with survival-improving effects of add-on VA in cancer patients are accumulating. Besides being effectively used in combination with radio-/and or chemotherapy [30,31,83], VA in combination with targeted therapy is associated with (i) a well-documented safety profile [48,49,50,51]; (ii) a significant reduction in adverse effects in cancer patients treated with monoclonal antibodies [85]; (iii) a reduction by approximately 50% in adverse event rates in patients with advanced or metastasised lung cancer treated with anti-PD-1 agents [49]; and (iv) improved ability to continue standard cancer therapy in patients treated with targeted therapies [51]. Thus, the synergistic association between combinatory PD-1/PD-L1 inhibitors and VA therapy in patients with advanced or metastasised NSCLC as observed in the present study joins the ranks of previous achievements.

### Limitations

The non-randomised nature of the present real-world data study limits our results. However, the confounding bias was reduced by applying adjusted multivariable logistic regression methods addressing potential confounders. It is worth noting that our findings suggest a correlation between add-on VA application and improved survival in advanced NSCLC patients receiving PD-1/PD-L1 inhibitors but do not establish causation. However, our data study mirrors the real-world application situation of PD-1/PD-L1 inhibitor therapy in NSCLC patients in standard clinical practice and is the first of its kind showing a positive survival association between combinatory PD-1/PD-L1 inhibitor and VA therapy. Owing to the real-world nature of our study, the occurrence of comorbidity (not distinguished between specific comorbidity variables) was documented. In the future, it will be necessary to detect further comorbidity conditions influencing combinational treatment response. Lastly, evaluated molecular markers in our cohort may not represent the current molecular marker panel as the RWE study started in 2015 and has been developed since. Further markers such as AXL, HER2, HER3, MERTK, VIM, and RAD18 or error-prone polymerases could be examined in further clinical trials evaluating resistance or compensatory pathways and how they are affected by the combinational PD-1/PD-L1 and add-on VA therapy in advanced or metastasised NSCLC patients. Prospective and randomised controlled trials are warranted to validate these findings and better understand the specific impact of combinatory PD-1/PD-L1 inhibitor plus VA therapy in lung cancer treatment.

## 5. Conclusions

The findings of our study suggest that the addition of VA to standard ICB therapy is associated with improved survival in patients with advanced or metastasised NSCLC irrespective of their age, metastasised state, performance status, lifestyle, or oncological treatment. The mechanisms may include synergistic modulations of the immune response by PD-1/PD-L1 inhibitors and VA. These findings may underline the clinical impact of add-on VA. However, our RWD data should be taken with caution due to the observational and non-randomised study design which only allows limited conclusions to be drawn. Prospective randomised trials are warranted.

## Figures and Tables

**Figure 1 cancers-16-01609-f001:**
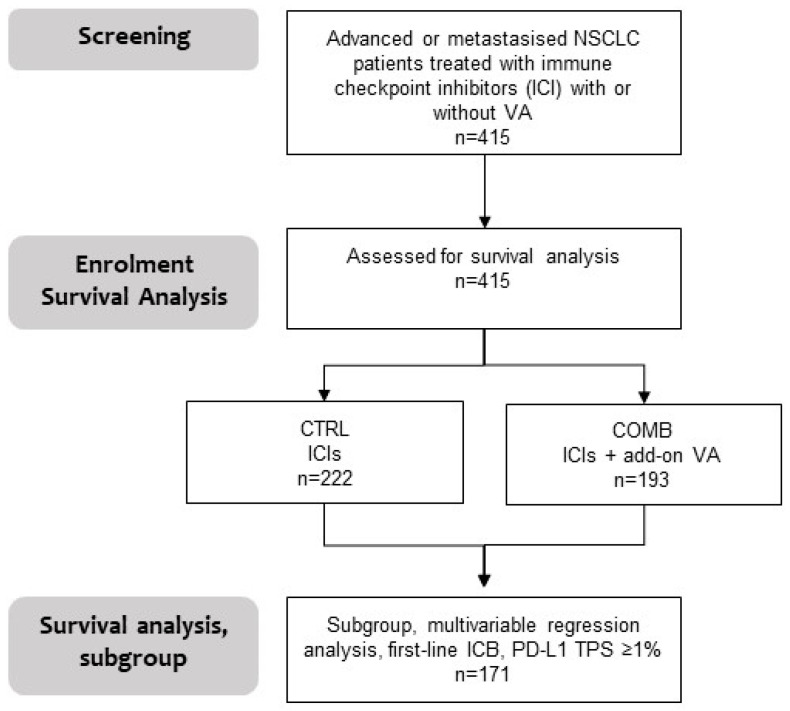
Flowchart of the study and analysis. A total of 415 participants with advanced or metastasised NSCLC treated with PD-1/PD-L1 inhibitors with (COMB) or without (CTRL) VA were included in the survival and adjusted multivariable regression analysis; furthermore, an adjusted multivariable regression analysis was performed for a subgroup of 171 patients with first-line immune checkpoint blockade (ICB) and PD-L1 status > 1%, TPS, tumour proportion status.

**Figure 2 cancers-16-01609-f002:**
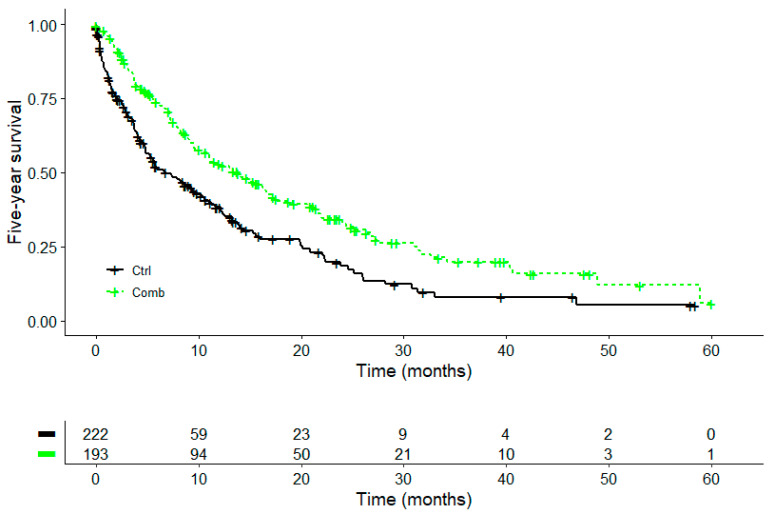
Five-year survival. Kaplan–Meier survival curves displaying 5-year survival in advanced or metastasised NSCLC patients treated with α-PD-1/PD-L1 without (CTRL, black line) or with add-on VA therapy (COMB, green line); ICB treatment in both groups consisted of first- and second-line PD-1/PD-L1 inhibitor therapy. ICB, immune checkpoint blockade; CRTL, PD-1/PD-L1 inhibitor treatment; COMB, α-PD-1/PD-L1 + add-on VA treatment; Log-rank test: χ^2^ = 17.5, *p* < 0.001.

**Figure 3 cancers-16-01609-f003:**
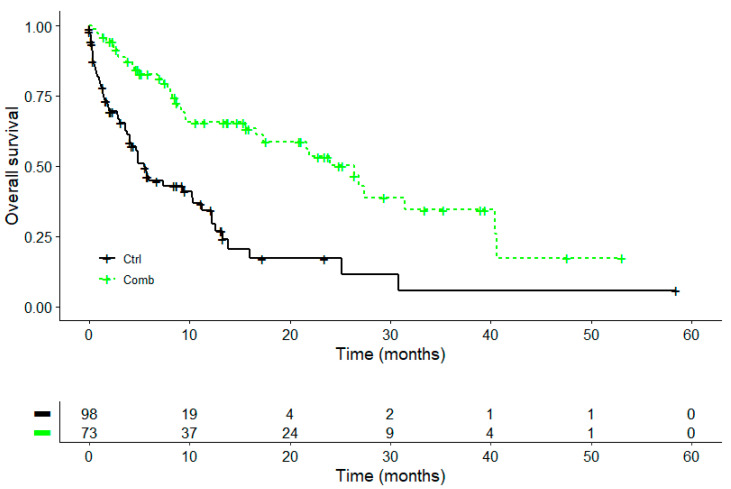
Subgroup with PD-L1 ≥ 1% advanced or metastasised NSCLC and first-line α-PD-1/PD-L1 without (CTRL, black line) or with add-on VA therapy (COMB, green line); *n* = 171. Kaplan–Meier survival curves displaying overall survival in patients; CRTL, α-PD-1/PD-L1 treatment; COMB, PD-1/PD-L1 inhibitor + add-on VA therapy; Log-rank test: χ^2^ = 24, *p* < 0.001.

**Figure 4 cancers-16-01609-f004:**
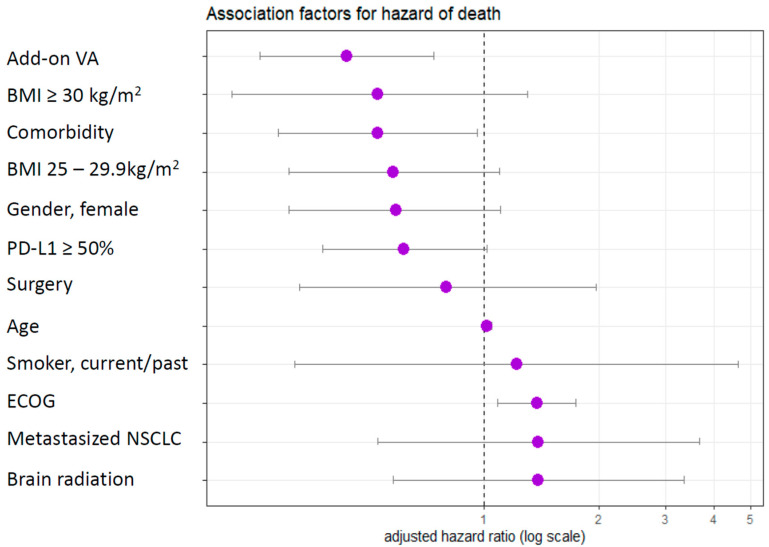
Subgroup with PD-L1-positive (>1%) advanced or metastasised NSCLC treated with first-line PD-1/PD-L1 inhibitors. Multivariate Cox proportional analysis of the association between add-on VA and adjusted hazard of death; *n* = 134. aHR, adjusted hazard ratio; VA, Viscum album L.; PD-1, programmed death protein 1; PD-L1, programmed death-ligand 1.

**Table 1 cancers-16-01609-t001:** Characteristics of patients.

	Total Group (*n* = 415)		CRTL (*n* = 222)		COMB (*n* = 193)		*p*-Value
	N	%	N	%	N	%	
Age (first diagnosis), median years (IQR)	68	(62–75)	69.5	(63.3–76.8)	67	(60.0–74.0)	0.552
Gender							0.616
Female	187	45.1	97	43.7	90	46.6	
Male	228	54.9	125	56.3	103	53.4	
Histology							0.214
non-squamous	271	65.3	141	63.5	130	67.4	
squamous	113	27.2	63	28.4	50	25.9	
NSCLC, NA	31	7.5	18	8.1	13	6.7	
Body mass index, kg/m^2^							0.11
<25	217	52.3	111	50.0	106	54.9	
25–29.9	133	32.0	65	29.3	68	36.2	
≥30	35	8.4	24	10.8	11	5.7	0.04 *
unknown	30	7.2	22	9.9	8	4.1	
ECOG							0.430
0	106	25.5	50	22.5	56	29	
1	177	42.7	100	45.5	77	39.9	
2	53	12.8	33	14.9	20	10.4	
≥3	38	9.2	20	9.0	18	9.3	
Smoker							0.10
current/past	319	76.9	166	74.8	153	79.3	
never	36	8.7	13	5.9	23	11.9	
unknown	60	14.5	43	19.4	17	8.8	
UICC stage							0.07
UICC stage III	90	21.7	40	18.0	50	25.9	
UICC stage IV	325	78.3	182	82.0	143	74.1	
Comorbidities							
Comorbidities, yes	334	80.5	175	78.8	159	82.4	0.43
Comorbidities, no	81	19.5	47	21.2	34	17.6	

Patient’s characteristics with advanced or metastasised NSCLC; percentages of sub-variables may not add to 100% due to calculation procedures where numbers were rounded. IQR, interquartile range; CRTL, patients treated with PD-1/PD-L1 inhibitors without VA; COMB, patients treated with PD-1/PD-L1 inhibitors and add-on VA; UICC, Union Internationale Contre le Cancer; ECOG, Eastern Cooperative Oncology Group; * chi-square for comparing obese (≥30 kg/m^2^) vs. normal weight (<25 kg/m^2^) and overweight (25–29.9 kg/m^2^).

**Table 2 cancers-16-01609-t002:** Molecular characteristics of patients’ non-small-cell lung cancer.

	Total Group (*n* = 415)		CRTL (*n* = 222)		COMB (*n* = 193)		*p*-Value
	N	%	N	%	N	%	
PD-L1 status							0.14
PD-L1 status known	341	82.2	185	83.3	156	80.8	
PD-L1 status, positive	239	57.6	137	61.7	102	52.8	0.08 *
PD-L1 status, negative	102	24.6	48	21.6	54	28.0	
PD-L1 status, positive TPS ≥ 50	122	29.4	73	32.9	49	25.4	0.15
ALK translocation							1
yes	1	0.2	1	0.5	0	0	
no	234	56.4	124	55.9	110	57.0	
EGFR (Exon 18–21) mutation							0.50
yes	13	3.1	6	2.7	7	3.6	
no	177	42.7	106	47.7	71	36.8	
ROS1-rearrangement							0.65
yes	0	0	0	0	0	0	
no	201	48.4	108	48.6	93	48.2	
BRAF V600E-mutation							1
yes	6	1.4	3	1.4	3	1.6	
no	205	49.4	114	51.4	91	47.2	

Molecular characteristics of NSCLC; percentages of sub-variables may not add to 100% because numbers were rounded. CRTL, patients treated with PD-1/PD-L1 inhibitors without VA; COMB, patients treated with PD-1/PD-L1 inhibitors and add-on VA; PD-L1, programmed death ligand; ALK, anaplastic lymphoma kinase; EGFR, epidermal growth factor receptor; ROS1, receptor tyrosine kinase encoded by the gene ROS1; BRAF, B-rapidly accelerated fibrosarcoma; TPS, tumour proportion score; * chi-square for comparing positive PD-L1 status to negative PD-L1 status.

**Table 3 cancers-16-01609-t003:** PD-L1 tumour proportion score of patients’ non-small-cell lung cancer.

	Total Group		CRTL (*n* = 185)		COMB (*n* = 156)		*p*-Value
	N	%	N	%	N	%	
PD-L1 status, known, *n*, %	341	100	185	100	156	100	0.46
PD-L1 status, TPS < 1%, *n*, %	102	29.9	48	25.9	54	34.6	
PD-L1 status, TPS 1–10%, *n*, %	70	20.5	43	23.2	27	18.6	
PD-L1 status, TPS 11–20%, *n*, %	13	3.8	9	4.9	4	2.8	
PD-L1 status, TPS 21–30%, *n*, %	16	4.7	9	4.9	7	4.8	
PD-L1 status, TPS 31–40%, *n*, %	6	1.8	3	1.6	3	2.1	
PD-L1 status, TPS 41–50%, *n*, %	10	2.9	6	3.2	4	2.8	
PD-L1 status, TPS 51–60%, *n*, %	28	8.2	17	9.2	9	6.2	
PD-L1 status, TPS 61–70%, *n*, %	22	6.5	14	7.6	8	5.5	
PD-L1 status, TPS 71–80%, *n*, %	21	6.2	12	6.5	9	6.2	
PD-L1 status, TPS 81–90%, *n*, %	32	9.4	16	8.6	16	11.0	
PD-L1 status, TPS 91–100%, *n*, %	11	3.2	8	4.3	3	2.1	

PD-L1 tumour proportion scores of NSCLC patients with known PD-L1 status (*n* = 341). CRTL, patients treated with PD-1/PD-L1 inhibitors without VA; COMB, patients treated with PD-1/PD-L1 inhibitors and add-on VA; PD-L1, programmed death ligand; TPS, tumour proportion score.

**Table 4 cancers-16-01609-t004:** Characterisation of oncological therapy.

	Total Group (*n* = 415)		CRTL (*n* = 222)		COMB (*n* = 193)		*p*-Value
	N	%	N	%	N	%	
Radiation, bone, *n*, %	38	9.2	17	7.7	21	10.9	0.317
Radiation, brain, *n*, %	41	9.9	17	7.7	24	12.4	0.133
Radiation, primary tumour, *n*, %	48	11.5	21	9.5	27	14.0	0.18
Radiation, abdomen, *n*, %	1	0.2	1	0.5	0	0	1
Radiation, other, *n*, %	10	2.4	4	1.8	6	3.1	0.57
Surgery, *n*, %	42	10.1	19	8.6	23	11.9	0.29
Chemotherapy, *n*, %	355	85.5	191	86	164	85	0.867
First-line immunotherapy, *n*, %	246	59.3	136	61.3	110	57.0	0.434
PD-L1/PD-1/CTL-A4 inhibitors							0.231
PD-L1 inhibitors, *n*, %	31	7.5	15	6.8	16	8.3	
PD-1 inhibitors, *n*, %	381	91.8	204	91.9	177	91.7	
CTL-A4 inhibitor, *n*, %	3	0.7	3	1.4	0	0	

Oncological therapy; *n*, number of patients; %, percent. CRTL, patients treated with PD-1/PD-L1 inhibitors without VA; COMB, patients treated with PD-1/PD-L1 inhibitors and add-on VA; PD-L1, programmed death ligand; PD-1, programmed cell death protein 1; CTL-A4, cytotoxic T-lymphocyte antigen 4.

**Table 5 cancers-16-01609-t005:** Median five-year survival in advanced or metastasised NSCLC patients.

	N	Events	Median [Months]	95% CI [Months]
NSCLC, CTRL	222	139	6.8	4.9–10.4
NSCLC, COMB	193	124	13.8	10.4–17.3
Log rank test χ^2^ = 17.5, *p* < 0.001				

Median five-year survival in patients with advanced or metastasised NSCLC according to treatment; *n* = 415.

**Table 6 cancers-16-01609-t006:** Multivariate Cox proportional analysis in advanced or metastasised NSCLC patients treated with PD-1/PD-L1 inhibitors.

	aHR	(95% CI)	*p*-Value
VA vs. non-VA	0.602	0.425–0.854	0.004 **
Age	1.003	0.984–1.023	0.773
Gender, female vs. male	0.588	0.401–0.862	0.006 **
BMI 25–29.9 kg/m^2^ vs. <25 kg/m^2^	0.623	0.424–0.917	0.020 *
BMI ≥ 30 kg/m^2^ vs. <25 kg/m^2^	0.473	0.260–0.864	0.015 *
ECOG status	1.298	1.090–1.545	0.003 **
Smoker, current/past vs. never	0.998	0.546–1.823	0.994
Comorbidities, yes vs. no	0.474	0.303–0.742	0.001 **
Surgery, yes vs. non	1.341	0.771–2.333	0.299
Stage IV UICC vs. stage III UICC	0.913	0.577–1.445	0.697
Brain radiation vs. no brain radiation	1.807	1.072–3.046	0.026 *
PD-L1 TPS 1–10%	0.949	0.607–1.484	0.819
PD-L1 TPS 11–20%	1.108	0.487–2.521	0.806
PD-L1 TPS 21–30%	0.880	0.442–1.751	0.716
PD-L1 TPS 31–40%	0.613	0.209–1.793	0.371
PD-L1 TPS 41–50%	2.726	0.6351–11.701	0.177
PD-L1 TPS 51–60%	0.839	0.421–1.672	0.619
PD-L1 TPS 61–70%	0.697	0.317–1.534	0.370
PD-L1 TPS 71–80%	0.680	0.300–1.540	0.355
PD-L1 TPS 81–90%	0.573	0.297–1.108	0.098
PD-L1 TPS 91–100%	0.761	0.261–2.218	0.062

Adjusted multivariate Cox proportional analysis of the association between add-on VA and adjusted hazard of death in advanced or metastasised NSCLC patients treated with PD-1/PD-L1 inhibitors; *n* = 248. aHR, adjusted hazard ratio; VA, Viscum album L.; BMI, body mass index; PD-L1, programmed death-ligand 1; TPS, tumour proportion score. *, *p*-value < 0.05; **, *p*-value < 0.01.

**Table 7 cancers-16-01609-t007:** Subgroup, treatment with first-line anti-PD-L1/PD-1 with aPD-L1 TPS score ≥1%. Median overall survival in advanced or metastasised NSCLC.

	N	Events	Median [Months]	95% CI [Months]
NSCLC, CTRL	98	56	5.4	4.1–11.3
NSCLC, COMB	73	34	26.4	16.6–NA
Log rank test χ^2^ = 24, *p* < 0.001				

Subgroup with PD-L1 TPS ≥ 1%, treated with first-line α-PD-L1/PD-1 without (CTRL) or with add-on VA (COMB); *n* = 171; NA, not applicable.

## Data Availability

All relevant data are included in this manuscript.

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
