# Peer review of "Patients with Advanced or Metastasised Non-Small-Cell Lung Cancer with Viscum album L. Therapy in Addition to PD-1/PD-L1 Blockade: A Real-World Data Study"

_cancers, 2024, doi:10.3390/cancers16081609_

Round 1

Reviewer 1 Report (Previous Reviewer 3)

Comments and Suggestions for Authors

The authors have appropriately addressed the comments I raised in my initial review and I recommend the paper to be acceptable for publication.

Author Response

Thank you for your careful review and your valuable comments. We have answered your query and have revised the manuscript according to your suggestions and comments.

We hope that we could sufficiently answer all of your questions and remarks and hope that our changes will meet your approval.

Response to Reviewer 1 Comments

The authors have appropriately addressed the comments I raised in my initial review and I recommend the paper to be acceptable for publication.

Answer to reviewer 1: Thank you very much again for your evaluation and we are delighted that it is now acceptable for you for publication in Cancers.

Reviewer 2 Report (New Reviewer)

Comments and Suggestions for Authors

Here are the comments: 

In a real-world study of advanced or metastasized non-small cell lung cancer (NSCLC) patients, combining Viscum album L. with PD-1/PD-L1 inhibitors resulted in a significant survival advantage compared to PD-1/PD-L1 inhibitor therapy alone. Patients in the combination group lived seven months longer on average than those in the control group (p < 0.001). Notably, in patients with PD-L1 positive tumors receiving first-line anti-PD-1/PD-L1 therapy, the addition of Viscum album L. reduced the adjusted hazard of death by 56% (p < 0.001). However, due to the study's observational and non-randomized design, prospective randomized trials are needed to confirm these findings and assess the safety and efficacy of this combination therapy further.  

Major Comments : 

  1. Authors should differentiate between EGFR mutations ( deletions, insertions or acquired mutations upon treatment) and then perform a detailed analysis.  
  2. What is the status of compensatory pathways (AXL, HER2 , HER3, MERTK, VIM, RAD18 and low-fidelity polymerases) Authors should monitor these statuses in patient samples and discuss. 
  3. How does inhibition of PD-L1/ immune response regulate AXL and low-fidelity? This can give a brief overview of the status of cancer cells if they relapse to form resistance.
  4. What is the status of EGFR  other driver mutations when treated with PD-L1 inhibitors?
  5. Does the status of expression of interleukin, cytokines and interferons affect the therapy response? What is the status of these in different mutation types?  And is it increased during metastasis? 
  • As the combination have been important in the field of lung cancer, this subject is indeed important. However, several important questions are not addressed in the manuscript. 
  • The study is above average to the other compared studies. Addressing the comments will enhance the quality of the study. Also Authors should mention recent advances in the subject. 
  • Methodology is well-addressed
  • References must be improved 

Author Response

Thank you for your careful review and your valuable comments. We have answered your query and have revised the manuscript according to your suggestions and comments.

We hope that we could sufficiently answer all of your questions and remarks and hope that our changes will meet your approval.

Response to Reviewer 2 Comments

In a real-world study of advanced or metastasized non-small cell lung cancer (NSCLC) patients, combining Viscum album L. with PD-1/PD-L1 inhibitors resulted in a significant survival advantage compared to PD-1/PD-L1 inhibitor therapy alone. Patients in the combination group lived seven months longer on average than those in the control group (p < 0.001). Notably, in patients with PD-L1 positive tumors receiving first-line anti-PD-1/PD-L1 therapy, the addition of Viscum album L. reduced the adjusted hazard of death by 56% (p < 0.001). However, due to the study's observational and non-randomized design, prospective randomized trials are needed to confirm these findings and assess the safety and efficacy of this combination therapy further.  

Major Comments: 

Reviewer 2, point 1: Authors should differentiate between EGFR mutations (deletions, insertions or acquired mutations upon treatment) and then perform a detailed analysis.  

Answer to point 1: Thank you very much for your review and evaluation. As suggested, we further analyzed the EGRF mutations and differentiated between insertion, deletion, duplication and point mutations. We inserted these information in the supplementary table 1A (see page 23) and in the text in the manuscript, see page 8 and lines 305-315:  “In the supplementary table 1A the different EGFR (Exon 18-21) mutations were further classified into EGFR insertion, deletion, point or duplication mutations. In total, two (0.5%) insertion mutations, five (1.2%) deletion mutations, six (1.4%) point mutations and two (0.5%) duplication mutations were documented for the thirteen patients with EGFR-mutations. The between-group comparison revealed no significant differences between both groups as to various mutation types, see supplementary table 1. Eight (1.9%) of the total observed EGFR-mutations were known to show sensitivity against tyrosine kinase inhibitors (TKIs) while two other mutations revealed no sensitivity against TKIs or no sensitivity against TKIs of the 1st, 2nd or 3rd generation. The between-group comparison revealed no significant differences between both groups as to sensitivity against TKIs, see supplementary table 1A.”

We also performed a detailed analysis and wrote in the results: “Adjusted multivariable cox regression analysis revealed a positive direction of the hazard of death for EGFR-mutant NSCLC (HR 1.07, 95%CI: 0.48- 2.39, p=0.87), see supplementary table 2A”, see page 8 and line 315-317. For supplementary table 2A, please see page 23 and page 24. We decided for a supplementary table as the number of observations in this analysis was too small and did not allow for a general assumptions. In the discussion we concluded: “We were unable to consider the association of EGFR mutations with overall survival in our study because only a small number of patients had EGFR mutations. However, adjusted multivariable cox regression analysis revealed a positive direction of the hazard of death, i.e. the probability of death is increased in the presence of common EGFR mutations.” See page 15 and lines 534 - 538. We hope you agree?

Reviewer 2, Point 2: What is the status of compensatory pathways (AXL, HER2 , HER3, MERTK, VIM, RAD18 and low-fidelity polymerases) Authors should monitor these statuses in patient samples and discuss. 

Answer to point 2: Thank you for your evaluation. Unfortunately, the focus of our study was to assess the overall survival of advanced or metastasized NSCLC patients registered in a real-world data registry which did not harbor data on biomarkers such as AXL, HER”, HER2, MERTK, VIM, RAD18 and low fidelity polymerases for the assessment of the immune profile of the patients to predict their response to treatment. Right now the molecular-pathological evaluation of HER2 and HER4 is already included in the national genome sequencing of primary NSCLC patients treated in certified lung cancer centers. However, in the Network Oncology registry the documentation of these data has only been started this year. The other markers such as AXL, HER3, MERTK, VIM, RAD18 and low-fidelity polymerase are not included yet in the national genome sequencing of primary NSCLC patients in Germany. We have therefore already mentioned in the limitations of our manuscript that “evaluated molecular markers in our cohort may not represent the current molecular marker panel as the RWE-study has started in 2015 and has been developed since”. We kindly would like to suggest to include in our limitations further the aspect of compensatory pathways so that it reads “Lastly, the evaluated molecular markers in our cohort may not represent the current molecular marker panel as the RWE-study has started in 2015 and has been developed since. Further markers such as AXL, HER2, HER3, MERTK, VIM, RAD18 or error-prone polymerases could be examined in further clinical trials evaluating resistance or compensatory pathways and how they are affected by the combinational PD-1/PD-L1 and add-on VA therapy in advanced or metastasized NSCLC patients.” See page 16 and line 623-628. We hope you agree?

Reviewer 2, point 3:  How does inhibition of PD-L1/ immune response regulate AXL and low-fidelity? This can give a brief overview of the status of cancer cells if they relapse to form resistance.

Answer to point 3: We included in our discussion a brief overview on resistance mechanisms of cancer cells in association with AXL and low-fidelity polymerases as well as on promising therapeutic strategy to overcome resistance to checkpoint inhibitor therapy in NSCLC, see discussion, page 15 lines 545– 566: “TKI treated NSCLCs with EGFR mutation often develop resistance which may limit therapy’s efficacy, mostly due to secondary mutations or compensatory pathways. The latter of the two adaptive mutability routes include the activation of pharmacologically sensitive endogenous genes that enable continuous error-prone DNA replication in order to tolerate cancer therapy. Blocking involved genes or proteins may overcome these resistance mechanisms (97, 98). One of these proteins is the tyrosine kinase receptor Axl which activates intrinsic mutators and helps tumor cells to evade tyrosine kinase inhibition (98). Treatment of NSCLC cell lines with EGFR TKI resulted in Axl overexpression, viability of tumor cells and inhibition of apopototic pathways. It was observed in in-vitro, in-vivo experiments and in patient samples that Axl mediated TKI-induction of error-prone DNA polymerases and acceleration of common EGFR resulted in drug resistance. In patient-derived xenografts this effect was reversed by anti-Axl therapy (97). The resistance to ICB is an emerging area in cancer therapy. Axl which has been correlated to immune suppression, resistance to immunity and lower response rates was shown to strongly associate with PD-1 expression of the tumor. When treated with PD-1 inhibitors, tumors with high Axl-expression revealed lower response rates and a trend toward shorter progression free survival (95). In addition, the activation of TAM receptors including Axl negatively modified the immune response leading to an immunosuppressive andpro-tumorigenic tumor microenvironment suggesting the combination of TAM receptor inhibitors and checkpoint inhibitors as long-term clinical therapeutic strategies in NSCLC (96, 98). Thus, combinational ICB strategies will in the future play a pivotal role to effectively reduce tumour cell survival and metastatic potential.

Reviewer 2, point 4: What is the status of EGFR other driver mutations when treated with PD-L1 inhibitors?

Answer to point 4: Thank you for this point. We felt that the new discussion paragraphs on EGFR mutations, their association with tumor immune responses and the brief introduction on resistance mechanisms and the involvement of AXL and its ability to induced low-fidelity polymerases seemed to give our discussion another focus than we intended. We therefore tended to abstain from further discussion on driver mutations, especially, as only a very small percentage had other driver mutations in our study. We hope you agree?

Reviewer 2, point 5: Does the status of expression of interleukin, cytokines and interferons affect the therapy response? What is the status of these in different mutation types?  And is it increased during metastasis? 

Answer to point 5: Thank you for your interesting point. We have now included in our discussion this aspect, see page 15 line 538-544. The efficacy of ICB can be influenced by the expression and activity of immune-related molecules which in turn are influenced by driver aberrations such as EGFR, ALK, KRAS or others. It has been shown that e.g. in EGFR-mutant lung cancer, the EGFR signaling pathway is activated leading likely to an uninflamed tumor microenvironment (93) due to the lowered activity of pro-inflammatory interleukines such as IL-6, IL-8 or TNF-α and due to the increased activity of immunosuppressive factors such as TGF-β resulting in tumor escape or immunosuppression. Thus, these patients do not benefit from an ICB therapy. We hope you agree?

Reviewer 2, point 6: As the combination have been important in the field of lung cancer, this subject is indeed important. However, several important questions are not addressed in the manuscript. The study is above average to the other compared studies. Addressing the comments will enhance the quality of the study. Also Authors should mention recent advances in the subject. Methodology is well-addressed. References must be improved 

Answer to point 6: Thank you for your evaluation and that “the study is above average to the other compared studies”. We have mentioned recent advances in the subject as suggested, page 2, lines 77 -  89: “Recent advances in ICB research and development have significantly improved options and outcomes for NSCLC patients including a) the approval as first-line combinational or mono-therapy in stage IV NSCLC without driver alterations depending on PD-L1 expression (78, 79, 85, 86, 87) or as an adjuvant therapy in non-metastatic NSCLC for certain conditions (80, 81) , b) the enhanced biomarker development including tumor mutational burden and microsatellite instability (82), c) the combination with other targeted, immuno-, chemo- or radiotherapy (87, 88, 89), d) the exploration and development of novel immune checkpoint inhibitors, multispecific antibodies and other immunomodulatory agents as well as the exploration of new immune regulation pathways (83, 84, 89). To maximize the efficacy and minimize toxicity of ICB or the development of resistance ongoing research is being continued for the optimization of immunotherapy approaches including personalized therapy strategies and research on immune evasion pathways, tumor microenvironment factors and host-immune response (89, 90, 91 92,).” We hope you agree? In addition, we have improved the references and included the following publications numbers 78 – 101 in line with the new formulated text passages.

Round 2

Reviewer 2 Report (New Reviewer)

Comments and Suggestions for Authors

The authors have addressed my comments. 

This manuscript is a resubmission of an earlier submission. The following is a list of the peer review reports and author responses from that submission.

Round 1

Reviewer 1 Report

Comments and Suggestions for Authors

Dear authors,

Congratulations for your article, "PD-1/PD-L1 Blockade in Combination with Viscum album L. Therapy is Associated with Improved Survival in Advanced or Metastasized NSCLC Patients, A Real-World Data Study". It is a significant contribution to the field of oncology, particularly in the treatment of non-small cell lung cancer (NSCLC). This real-world data study investigates the efficacy of adding Viscum album L. (VA), a type of mistletoe extract, to the standard PD-1/PD-L1 inhibitor therapy in patients with advanced or metastasized NSCLC.

The study is well-structured and comprehensive, presenting clear and detailed methodologies, statistical analyses, and results. Data from the accredited national Network Oncology registry was used, covering 415 patients treated either with PD-1/PD-L1 inhibitors alone (control group) or in combination with VA (combination group). The finding that the combination group exhibited a median survival of 13.8 months compared to 6.8 months in the control group is particularly noteworthy. This significant improvement in survival rates underscores the potential of VA as an adjunct therapy in NSCLC treatment.

The study also stands out for its thorough analysis of various patient characteristics, ensuring a balanced comparison between the two groups. The use of multivariate Cox proportional hazard analysis to adjust for confounding factors adds to the credibility of the results. Moreover, the exploration of the potential mechanisms behind VA's efficacy, particularly its immunomodulatory properties, offers valuable insights for future research.

However, the study's real-world design, while reflective of clinical practice, does have inherent limitations, including the non-randomized nature of the treatment groups. This aspect is well-acknowledged, recommending prospective randomized trials for more definitive conclusions.

Overall, the study is an important addition to the ongoing research in cancer immunotherapy. It opens new avenues for integrating traditional herbal medicines like VA in modern oncological treatments, potentially enhancing patient outcomes in advanced NSCLC. The study is well-executed and presents a compelling case for the combination of PD-1/PD-L1 inhibitors and VA, setting the stage for further clinical trials to validate and understand the full potential of this therapeutic approach.

Author Response

Response to Reviewer 1

Thank you for your careful review and your valuable comments. We have answered your query and have revised the manuscript according to your suggestions and comments.

In addition, as suggested by the academic editor, we have made several changes in the title, summary, abstract and manuscript text to minimize overlaps with already published sources. Last but not least, we included a new figure in the text (figure 4).

We hope that we could sufficiently answer all of your questions and remarks and hope that our changes will meet your approval.

Reviewer 1: Congratulations for your article, "PD-1/PD-L1 Blockade in Combination with Viscum album L. Therapy is Associated with Improved Survival in Advanced or Metastasized NSCLC Patients, A Real-World Data Study". It is a significant contribution to the field of oncology, particularly in the treatment of non-small cell lung cancer (NSCLC). This real-world data study investigates the efficacy of adding Viscum album L. (VA), a type of mistletoe extract, to the standard PD-1/PD-L1 inhibitor therapy in patients with advanced or metastasized NSCLC. The study is well-structured and comprehensive, presenting clear and detailed methodologies, statistical analyses, and results. Data from the accredited national Network Oncology registry was used, covering 415 patients treated either with PD-1/PD-L1 inhibitors alone (control group) or in combination with VA (combination group). The finding that the combination group exhibited a median survival of 13.8 months compared to 6.8 months in the control group is particularly noteworthy. This significant improvement in survival rates underscores the potential of VA as an adjunct therapy in NSCLC treatment. The study also stands out for its thorough analysis of various patient characteristics, ensuring a balanced comparison between the two groups. The use of multivariate Cox proportional hazard analysis to adjust for confounding factors adds to the credibility of the results. Moreover, the exploration of the potential mechanisms behind VA's efficacy, particularly its immunomodulatory properties, offers valuable insights for future research. However, the study's real-world design, while reflective of clinical practice, does have inherent limitations, including the non-randomized nature of the treatment groups. This aspect is well-acknowledged, recommending prospective randomized trials for more definitive conclusions. Overall, the study is an important addition to the ongoing research in cancer immunotherapy. It opens new avenues for integrating traditional herbal medicines like VA in modern oncological treatments, potentially enhancing patient outcomes in advanced NSCLC. The study is well-executed and presents a compelling case for the combination of PD-1/PD-L1 inhibitors and VA, setting the stage for further clinical trials to validate and understand the full potential of this therapeutic approach.

Answer to reviewer 1: Thank you very much for your evaluation that our study “is well-executed and presents a compelling case for the combination of PD-1/PD-L1 inhibitors and VA” and that is adding “a significant contribution to the oncological field”.

Reviewer 2 Report

Comments and Suggestions for Authors

The study aimed to evaluate overall survival of advanced or metastasized NSCLC patients receiving Immune checkpoint blockade (ICB) as outlined above with or without VA. 
This is not a novel as plant extract preparations from European mistletoe (Viscum album L.) have been studied as supportive supplement of cancer treatment in clinical setting particularly in Western Europe for more than half of the century. Several studies have been published with very minor differences. 
The study design is not appropriate as the outcome is entirely dependent on sample selection criteria. 
Several important clinical parameters related to patient lifestyle and clinical-pathology are completely ignored by the authors. Analytical approach is very weak. No univariate analysis provided of the multivariate analysis parameters to find the changes from independent to combined analysis. 

Author Response

Response to Reviewer 2

Thank you for your careful review and your valuable comments. We have answered your query and have revised the manuscript according to your suggestions and comments.

In addition, as suggested by the academic editor, we have made several changes in the title, summary, abstract and manuscript text to minimize overlaps with already published sources. Last but not least, we included a new figure in the text (figure 4).

We hope that we could sufficiently answer all of your questions and remarks and hope that our changes will meet your approval.

Reviewer 2, point 1: The study aimed to evaluate overall survival of advanced or metastasized NSCLC patients receiving Immune checkpoint blockade (ICB) as outlined above with or without VA. This is not a novel as plant extract preparations from European mistletoe (Viscum album L.) have been studied as supportive supplement of cancer treatment in clinical setting particularly in Western Europe for more than half of the century. Several studies have been published with very minor differences. 

Answer to point 1: Thank you very much for your review. We do not agree with your comment on the “very minor differences” for the supportive care with VA. There have been already more than 290 published studies with VA including more than 30 randomized controlled trials and systematic reviews and meta-analyses with add-on VA in the oncological setting. Current systematic reviews and meta-analyses have shown a medium effect size (d=0,61 (95% CI 0,41-0,81; p<0,0001) for add-on VA in the improvement of health-related quality of life in oncological patients (Loef et al. 2020) and a moderate effect (mean difference of -0,48, 95% CI: -0,82 to -0,14, p=0.006) in the improvement of cancer-related fatigue (Pelzer et al. 2020). Moreover, recent systematic reviews and meta-analyses revealed significant improvements in the survival of oncological patients treated with add-on VA therapy with a reduction of hazard ranging between 19% (HR 0.81, 95%CI: 0.69-0.95, p=0.01) (Loef et al. 2022) to 41% (HR 0.59, 95%CI: 0.53 – 0.65, p<0.0001) (Ostermann et al. 2020). In order to improve the background information in our introduction as suggested, we elaborated these aspects in more detail on page 3 line 137 to 145  – we hope you agree. We have also, as suggested by another reviewer, included further variables in the patient’s characteristics (comorbidity, BMI, smoking status), see table 1 on page 6, and observed that the significant association of add-on VA and survival outcome remained, see table 5 on page 7 and figure 4 on page 13 and results on page 12, line 427 to 429.

Pelzer F, Loef M, Martin DD, Baumgartner S. Cancer-related fatigue in patients treated with mistletoe extracts: a systematic review and meta-analysis. Support Care Cancer. 2022 Aug;30(8):6405-6418. doi: 10.1007/s00520-022-06921-x. Epub 2022 Mar 3. PMID: 35239008; PMCID: PMC9213316.

Loef M, Walach H. Survival of Cancer Patients Treated with Non-Fermented Mistletoe Extract: A Systematic Review and Meta-Analysis. Integr Cancer Ther. 2022 Jan-Dec;21:15347354221133561. doi: 10.1177/15347354221133561. PMID: 36324298; PMCID: PMC9634211.

Loef M, Walach H. Quality of life in cancer patients treated with mistletoe: a systematic review and meta-analysis. BMC Complement Med Ther. 2020 Jul 20;20(1):227. doi: 10.1186/s12906-020-03013-3. PMID: 32690087; PMCID: PMC7370416.

Ostermann T, Appelbaum S, Poier D, Boehm K, Raak C, Büssing A. A Systematic Review and Meta-Analysis on the Survival of Cancer Patients Treated with a Fermented Viscum album L. Extract (Iscador): An Update of Findings. Complement Med Res. 2020;27(4):260-271. English. doi: 10.1159/000505202. Epub 2020 Jan 10. PMID: 31927541.

Reviewer 2, point 2: The study design is not appropriate as the outcome is entirely dependent on sample selection criteria. 

Answer to point 2: Thank you for your evaluation. Your statement aims towards the assumption that in general the patients of our study that did not receive add-on VA could have been in a worse condition and would therefore have had a worse survival outcome compared to patients receiving the add-on VA. However, we could show in our study that patient’s characteristics were balanced between both groups. In addition, every patient enrolled in the study was in a good enough condition to receive immunotherapy (or even sometimes chemotherapy) with its known toxicity profile. Furthermore, we applied a multivariate analysis to control for confounders and took account for the combined influence of several factors including age, ECOG status, comorbidities, nutritional status, other oncological treatment, histological subtype etc. Following this, we still observed the significant association of add-on VA with improved survival outcome. In addition, our real-world data study design does not establish a causation and describes rather an association with improved survival in advanced or metastasized NSCLC patients treated with the combined PD-1/PD-L1 plus add-on VA therapy. Therefore, the wording strictly adhered to ,associations, and not ,causations, throughout the whole manuscript, see page 1: line 33, 38, and 43; page 2: line 48, page 3: line 139, 146, 147; page 4: line 170; page 5: line 232; page 10: line 376 and 377; page 10: line 327; page 11: line 328; page 13: line 460, 466, 475, page 14: line 516; page 15: line 544, 546, 550, 558, 578, 585 and page 16: line 601 and 612. Furthermore, we have already pointed towards these aspects in our limitations: “our findings suggest a correlation between add-on VA application and improved survival in advanced NSCLC patients receiving PD-1/PD-L1 inhibitors but do not establish causation.“…”prospective and randomized are warranted to validate these findings and to better understand the specific impact of combinatory PD-1/PD-L1 inhibitor plus VA therapy in lung cancer treatment” (page 16, line 596-599 and line 607-609). We hope you agree with our explanation.

Reviewer 2, point 3: Several important clinical parameters related to patients lifestyle and clinical-pathology are completely ignored by the authors. 

Answer to point 3: We have now included further clinical parameters in our analyses related to patient’s lifestyle and clinical pathology, to examine their contribution to the outcome, see table 5 on page 11, figure 4 on page 13 as well as results on page 10 and line 375-377 and on page 12 and line 429-432. We observed that a) these clinical parameters were as well balanced between both groups (univariate statistics, baseline characteristics) and b) that the significant association of add-on VA with improved survival remained (adjusted multivariate analysis). We hope you agree with this.

Reviewer 2, point 4: Analytical approach is very weak. No univariate analysis provided of the multivariate analysis parameters to find the changes from independent to combined analysis. 

Answer to point 4: Thank you for your evaluation, however we do not fully agree. Firstly, we did provide univariate analyses in the beginning to detect correlations of patients’ baseline variables with treatment (baseline analysis, table 1) or associations of survival outcome with treatment (Kaplan-Meier survival analyses, figure 2 and 3). In addition, we performed multivariate analyses to study the impact of the add-on VA treatment on the outcome (hazard of death) while controlling at the same time for any confounding variable (table 5 and now new figure 4).  While univariate analyses examine only one variable at a time, multivariable analyses provide a more accurate picture of the relationship among variables compared to univariate analysis. Multivariate analysis provides a more realistic representation of complex relationships of multiple variables that might be missed in univariate analyses. We therefore applied multivariate Cox regression analysis to control for confounding variables such as age, gender, histology, lifestyle or treatment variables. Thus, multivariate analyses can help identify the most significant factors outcomes, offering valuable guidance for interventions. We hope you agree. As to our chosen analytical approach the first head-to-head comparison study between real-world data and randomized controlled trials as to immunotherapy in stage IV NSCLC pointed towards the importance of evaluating differences between the efficacy of systemic therapies seen in RCTs and the effectiveness in real-world clinical practice (Cramer et al. 2021). This so-called efficacy-effectiveness gap has been shown for the overall survival for first-line pembrolizumab therapy in stage IV NSCLC patients. This is partly due to higher number of included patients with performance status ≥ 2 resulting in shorter OS with a probability of hazard of 1.55 (95% CI 1.07–2.25) in real-world first-line pembrolizumab therapy (Cramer et al. 2021). Thus, our analytical RWE approach may add important value to detect the real-world effectiveness of immunotherapy and combinational approaches.

Cramer-van der Welle CM, Verschueren MV, Tonn M, Peters BJM, Schramel FMNH, Klungel OH, Groen HJM, van de Garde EMW; Santeon NSCLC Study Group. Real-world outcomes versus clinical trial results of immunotherapy in stage IV non-small cell lung cancer (NSCLC) in the Netherlands. Sci Rep. 2021 Mar 18;11(1):6306. doi: 10.1038/s41598-021-85696-3. PMID: 33737641; PMCID: PMC7973789.Formularbeginn

Reviewer 3 Report

Comments and Suggestions for Authors

The authors present a retrospective analysis to study the impact of the addition of VA to PD-1/PDL-1 blockade inhibitors in the treatment NSCLC.  The appropriate include different stages of NSCLC as well as other demographics.

I would suggest addressing several additional points:

1. is there any association between which PD-1/PDL-1 blockade inhibitor was used and the impact of VA treatment?

2. was the addition of VA administration, in terms of timing, etc, the same in the selected studies?

3. were there any additional co-morbid conditions in the patients that might have impacted response to treatment?

Author Response

Response to Reviewer 3

Thank you for your careful review and your valuable comments. We have answered your query and have revised the manuscript according to your suggestions and comments.

In addition, as suggested by the academic editor, we have made several changes in the title, summary, abstract and manuscript text to minimize overlaps with already published sources. Last but not least, we included a new figure in the text (figure 4).

We hope that we could sufficiently answer all of your questions and remarks and hope that our changes will meet your approval.

Reviewer 3, point 1: The authors present a retrospective analysis to study the impact of the addition of VA to PD-1/PDL-1 blockade inhibitors in the treatment NSCLC.  The appropriate include different stages of NSCLC as well as other demographics. I would suggest addressing several additional points: is there any association between which PD-1/PDL-1 blockade inhibitor was used and the impact of VA treatment?

Answer to point 1: Thank you for this suggestion. We observed in our study that add-on VA was primarily associated with the reduced adjusted hazard of death in first-line PD-1 inhibitor treated patients with PD-L1 positive NSCLC as more than 98% of patients in this subgroup received a PD-1 inhibitor. This interesting association with PD-1 inhibition will be part of another research in our group, therefore we would like to refrain from disclosing more details on these early results in the present manuscript. Additionally performed randomized controlled trials in the near future will be mandatory to unveil the synergistic mechanisms of the combined treatment of PD-1 inhibition and add-on VA. We hope you agree.

Reviewer 3, point 2. was the addition of VA administration, in terms of timing, etc, the same in the selected studies?

Answer to point 2:

Thank you for this question. We have now included a passage on the timing of VA administration alongside ICB, page 15, line 561-573, we hope you agree:

In the combinational setting of the present study, ICB with add-on VA was started in the first-line setting in median 24 days after NSCLC diagnosis. Until now, no systematic research on the combinational ICB plus VA therapy and timing exists in literature with our data providing first information so far. For other combinations including chemotherapy and add-on VA, findings of a RCT reported an application start of the combination of one week after surgery with a duration of 23 weeks in patients with gastric cancer (75). Another study, a real-world data study, revealed an effective application duration of add-on VA at a minimum of four weeks alongside chemotherapy in advanced or metastasized NSCLC patients and the adjusted hazard was further reduced when add-on VA duration was prolonged to greater sixteen weeks (10). Add-on VA can be applied directly following tumor diagnosis, in a neo-adjuvant setting or in an adjuvant setting during chemo- and/or radio-chemotherapy (74) helping to improve patient’s health-related quality of life by reducing adverse effects of standard-oncological therapies (38, 76, 23).”

Reviewer 3, point 3. were there any additional co-morbid conditions in the patients that might have impacted response to treatment?

Answer to point 3: Thank you for this question. With regards to gender, lifestyle or comorbid conditions, we observed in the general study group (adjusted multivariate regression analysis) that a comorbid status, an overweight or obese BMI or a female gender were associated with a reduced adjusted hazard of death while an increased ECOG status was associated with an increased risk. However, in the subgroup (PD-L1 status positive NSCLC, treatment with first-line PD-1/PD-L1 therapy) only comorbidity (reduced hazard, aHR 0.53, 95%CI: 0.29-0.96) and ECOG status (increased hazard, aHR, 1.38, 95%CI: 1.09-1.74), remained. We have added this information in the results, see page 12 line 430 to 432 and discussed our findings, see page 15 and line 544 - 560. Owing to the real-world character of our study, the occurrence of comorbidity (not distinguished between specific comorbidity variables) was documented. In the future it will be mandatory to detect further co-morbidity conditions influencing combinational treatment response. We have added this last sentence in the limitations, page 16 and line 602 - 605. We hope you agree.

Round 2

Reviewer 2 Report

Comments and Suggestions for Authors

The paper lacks novelty and the data is not strong enough.